# Baseline Dysregulation in B, T, and NK Cells in COVID-19 Predicts Increased Late Mortality but Not Long-COVID Symptoms: Results from a Single-Center Observational Study

**DOI:** 10.3390/v17101400

**Published:** 2025-10-21

**Authors:** Aleksandra Matyja-Bednarczyk, Radosław Dziedzic, Anna Drynda, Ada Gradzikiewicz, Monika Bociąga-Jasik, Krzysztof Wójcik, Sabina Lichołai, Karolina Górka, Natalia Celejewska-Wójcik, Tomasz Stachura, Kamil Polok, Lech Zaręba, Teresa Iwaniec, Krzysztof Sładek, Stanisława Bazan-Socha

**Affiliations:** 1Jagiellonian University Medical College, Prof A. Szczeklik 2nd Chair of Internal Medicine, Department of Allergy, Autoimmunization and Hypercoagulation, Jakubowskiego 2, 30-668 Kraków, Poland; aleksandra.matyja-bednarczyk@uj.edu.pl (A.M.-B.); krzysztof.wojcik@uj.edu.pl (K.W.); 2Jagiellonian University Medical College, Doctoral School of Medical and Health Sciences, św. Łazarza 16, 31-530 Kraków, Poland; radoslaw.dziedzic@doctoral.uj.edu.pl (R.D.); anna.drynda@doctoral.uj.edu.pl (A.D.); 3University Hospital in Kraków, Department of Rheumatology, Immunology and Internal Medicine, Jakubowskiego 2, 30-688 Kraków, Poland; adagradzikiewicz97@gmail.com; 4Jagiellonian University Medical College, Department of Infectious and Tropical Diseases, Jakubowskiego 2, 30-668 Kraków, Poland; monika.bociaga-jasik@uj.edu.pl; 5Jagiellonian University Medical College, Prof A. Szczeklik 2nd Chair of Internal Medicine, Department of Molecular Biology and Clinical Genetics, Skawińska 8, 31-066 Kraków, Poland; sabina.licholai@uj.edu.pl; 6Jagiellonian University Medical College, Prof A. Szczeklik 2nd Chair of Internal Medicine, Department of Pulmonology, Jakubowskiego 2, 30-668 Kraków, Poland; karolina.gorka@uj.edu.pl (K.G.); natalia.celejewska@uj.edu.pl (N.C.-W.); tomasz.stachura@uj.edu.pl (T.S.); krzysztof.sladek@uj.edu.pl (K.S.); 7Jagiellonian University Medical College, Department of Intensive Care and Perioperative Medicine, Jakubowskiego 2, 30-668 Kraków, Poland; kamil.polok@uj.edu.pl; 8University of Rzeszów, Faculty of Exact and Technical Science, Institute of Computer Science, Pigonia 1, 35-310 Rzeszów, Poland; lzareba@ur.edu.pl; 9Jagiellonian University Medical College, Department of Hematology, Jakubowskiego 2, 30-688 Kraków, Poland; teresa.iwaniec@uj.edu.pl

**Keywords:** SARS-CoV-2, COVID-19, long-COVID, lymphopenia, lymphocyte subpopulations

## Abstract

The SARS-CoV-2 pandemic presents a broad clinical spectrum from asymptomatic cases to severe respiratory failure with high mortality. Severe COVID-19 is characterized by immune dysregulation, including lymphopenia and alterations in the counts of T, B, and NK cells in peripheral blood. Due to the limited data on long-term outcomes related to immune dysregulation, we aimed to analyze immunologic features at baseline in severe and mild COVID-19 cases and assess follow-up characteristics associated with later mortality and long-COVID signs. We included adult patients consecutively hospitalized with COVID-19 between June and November 2020 at the University Hospital in Kraków, corresponding to the first and second waves of COVID-19 in Poland. We enrolled only those who had been thoroughly assessed in terms of clinic and laboratory data, including immunological workups, and survived the acute phase of the disease. In 2025, between February and April (median time of follow-up: 54 months), we conducted a telephone questionnaire on long-COVID symptoms among survivors who had given their consent. Statistical analyses were performed to compare groups with severe and mild disease in terms of dysregulation in lymphocyte subpopulations and the follow-up outcomes. The study included 103 COVID-19 patients, comprising 53 severe (based on the need for at least high-flow nasal oxygen therapy) and 50 mild cases, with no differences in age, sex, and body mass index. Severe COVID-19 patients compared to mild cases had lower CD3+ T cells (count and percentage), CD4+ T cells (count and percentage), CD8+ T cells (count), and NK cells (count), but higher CD19+ B cells (percentage) at baseline (*p* < 0.05, all). At the time of follow-up, we evaluated 80 patients (77.7% of the baseline participants), with 23 (22.3%) patients lost to follow-up. Among patients analyzed in the follow-up, 23 (28.8%) had died, and 29 of the 57 survivors (50.9%) reported persistent long-COVID symptoms. Patients who died had significantly lower baseline counts of CD3+ T cells (377 vs. 655 cells/µL), CD4+ T cells (224 vs. 372 cells/µL), CD8+ T cells (113 vs. 188 cells/µL), and NK cells (118 vs. 157 cells/µL) compared to survivors (*p* < 0.05, all). Notably, the percentage of CD19+ B cells was higher in deceased individuals (19.2% vs. 13.5%; *p* = 0.049). In contrast, we did not document differences in baseline immunological data among survivors with and without long-COVID signs. Our study suggests that dysregulation in lymphocyte subpopulations during the COVID-19 acute phase may be associated with increased late mortality, but not with the persistence of long-COVID symptoms.

## 1. Introduction

The emergence of the novel coronavirus, severe acute respiratory syndrome coronavirus 2 (SARS-CoV-2), and its associated coronavirus disease 2019 (COVID-19) has caused a global health crisis [1]. Since its initial identification in December 2019 in Wuhan, China [2], the virus has rapidly spread across the world, leading to widespread morbidity and mortality [1]. The clinical spectrum of COVID-19 varies widely, with asymptomatic carriers comprising a significant proportion of infected individuals and further ranging from mild common cold to severe acute respiratory distress syndrome (ARDS) and death [2,3]. SARS-CoV-2 infection can activate both innate and adaptive immune responses, leading to severe and potentially life-threatening inflammation and tissue damage [4]. Symptomatic cases typically present with fever, cough, and dyspnea, while severe manifestations comprise pneumonia, acute respiratory distress syndrome, multi-organ failure, and death [5]. Comorbidities have emerged as significant determinants of COVID-19 clinical outcomes, shaping the disease course and prognosis for affected individuals. Patients with pre-existing conditions such as hypertension, diabetes mellitus, chronic kidney disease, cardiovascular disease, chronic respiratory diseases, malignancy, and obesity have been shown to experience more severe manifestations of COVID-19, including death [6]. Interestingly, various biochemical markers, for instance C-reactive protein (CRP) [7], procalcitonin (PCT) [8], lactate dehydrogenase (LDH) [9], D-dimer [10], and interleukin(IL)-6 [11], have been implicated as indicators of systemic inflammation and organ damage in this disease.

Dysregulation of lymphocyte subpopulations has emerged as a hallmark feature of COVID-19 pathogenesis [12,13]. Studies have reported both overall lymphopenia and alterations in various lymphocyte subsets, including T and B cells, and natural killer (NK) cells, in response to SARS-CoV-2 infection [12,13]. In patients with severe COVID-19, leukopenia and lymphopenia, accompanied by a reduction in the percentages of CD4+ and CD8+ T cells, are common features. Several mechanisms are speculated to contribute to lymphopenia in COVID-19, including the effects of the cytokine storm and retention of T cells in lymphoid organs and on the endothelial surface. A better understanding of the immunopathology of COVID-19 may contribute to understanding its later complications, such as the development of long-COVID. However, there is a lack of studies on COVID-19 patients in the context of a disturbed immune system during acute infection and its later impact on long-term outcomes, including mortality and the so-called long-COVID [14]. Moreover, in a recent study, Fresi et al. [15] explored whether machine learning could identify distinct phenotypes in long-COVID patients.

However, despite advanced methods, no clear symptom clusters were found, highlighting the high heterogeneity of the long-COVID group. There is a lack of long-term follow-up data addressing whether early immunological disturbances after acute COVID-19 have implications for the development of long-COVID. Thus, further research integrating immunological profiles with clinical data might be beneficial to uncover meaningful patterns and improve personalized care strategies, especially for long-COVID patients. In this study, we aimed to investigate the clinical, laboratory, and immunological characteristics of severe and mild COVID-19 patients and determine whether baseline immune-related dysregulations are associated with mortality and long-COVID symptoms in a long-term follow-up.

## 2. Patients and Methods

### 2.1. Study Population

We included in the study 103 symptomatic adult Caucasian patients with COVID-19 who were consecutively admitted to the Department of Pulmonology and Allergology at the University Hospital in Kraków, Poland, from June 2020 to November 2020, corresponding to the first and second waves of COVID-19 in Poland. We enrolled only those who had thoroughly assessed detailed data on disease severity and laboratory results, including immunological workups, and had survived the acute phase of the disease. The diagnosis was established based on clinical symptoms and positive results for SARS-CoV-2 in nasal swabs obtained via real-time reverse transcription polymerase chain reaction testing.

On admission, patients were assessed for a severe pneumonia course by the Modified Early Warning Score (MEWS) and for pulmonary embolism using Wells and modified Geneva scores. Additionally, we assessed the Pulmonary Embolism Severity Index (PESI) in all patients. Blood samples for laboratory tests were taken on admission to the hospital and before any other treatment was introduced. After that, patients were followed up for the disease course until discharge. We collected data on demographic factors, including age, sex, and body mass index, as well as clinical characteristics, concomitant diseases, and clinical symptoms of COVID-19. Additionally, we recorded parameters during hospitalization (e.g., systolic and diastolic blood pressure, heart rate, oxygen saturation, and temperature), as well as treatment patterns.

We excluded patients from the study with active cancer and congestive heart failure (WHO class III/IV). Individuals with liver injury and kidney insufficiency were eligible if they had been diagnosed as related to COVID-19 infection and had not presented with these conditions in the preceding six months. Liver injury was defined as an increase in serum alanine transaminase more than twice the upper limit of the normal range. Kidney insufficiency was defined as an estimated glomerular filtration rate (eGFR) of less than 60 mL/min/1.73 m^2^.

For further analysis, we divided the cohort of 103 COVID-19 patients into two groups. The first consisted of patients with a severe course of COVID-19 (*n* = 53) defined as requiring at least high-flow nasal oxygen therapy. The second group comprised the remaining COVID-19 patients (*n* = 50) with a mild course of disease, who did not require active oxygen therapy.

The study was approved by the Bioethics Committee of the Jagiellonian University Medical College (No. 1072.6120.333.2020). The study procedures were carried out under the ethical guidelines of the Declaration of Helsinki. All subjects were given a thorough description of the methods used in the study, along with the safety protocol, before obtaining informed consent to participate.

### 2.2. Basic Laboratory Tests

Routine laboratory techniques were performed to evaluate complete blood cell count, serum glucose, alanine and aspartate transaminase, gamma-glutamyltransferase, lactate dehydrogenase, bilirubin, urea, creatinine, N-terminal pro-B-type natriuretic peptide (NT-proBNP), creatine kinase, myoglobin, troponin I, procalcitonin, and C-reactive protein (CRP). The eGFR was estimated by the MDRD formula. Circulating IL-6 was measured using an electrochemiluminescent technique on a COBAS E601 analyzer (Roche, Meylan, France).

### 2.3. Coagulation Tests and Assessment of Antiphospholipid Antibodies

Prothrombin time (PT), activated partial thromboplastin time (APTT), and fibrinogen were determined by routine laboratory assays (Siemens, Marburg, Germany). The activities of antithrombin and protein C were measured using chromogenic methods (Innovance Antithrombin, Berichrom Protein C; Siemens, Marburg, Germany). Free protein S and D-dimer were assessed by turbidimetric assay (Innovance Free protein S and Innovance D-dimer; Siemens, Marburg, Germany). Activated protein C (APC) resistance was analyzed by a coagulometric test (ProCAcR Siemens, Marburg, Germany). Commercially available immunoenzymatic assays were applied to determine anticardiolipin and anti-β2-glycoprotein I antibodies (QUANTA Lite^®^ aCL and aβ2GPI, Inova Diagnostics, San Diego, CA, USA).

### 2.4. Evaluation of Immunological Parameters

Fasting whole blood was collected aseptically from every patient by venipuncture into ethylenediamine tetraacetic acid (EDTA) collection tubes for the quantification of the main lymphocyte subpopulations. Whole blood was incubated with BD Multitest 6-color TBNK reagent and then lysed with BD FACS™ lysing solution. Lymphocyte subpopulations were acquired and analyzed with BD FACSCanto clinical software. The BD Multitest 6-color TBNK reagent contains the following antibodies to identify and count different lymphocyte subsets: CD3 FITC was used for T lymphocyte identification, CD16 and CD56 PE for NK lymphocyte identification, CD45 PerCP-Cy™5.5 for lymphocyte population identification, CD4 PE-Cy™7 for T-helper/inducer lymphocyte identification and CD19 APC B lymphocyte identification, and CD8 APC-Cy7 for inhibitory/toxic T lymphocyte subset identification.

### 2.5. Follow-Up Analysis

In 2025, between February and April (median time of follow-up: 54 months), we conducted a telephone questionnaire on long-COVID symptoms among survivors who had given their consent (for details of the survey questions, see the Appendix A). Patients or their families were contacted using the phone number obtained from hospital records to determine the survival status of the study participants. For those who were alive and consented to further participation, a detailed questionnaire was administered to assess the presence and characteristics of persistent symptoms following COVID-19 infection. These symptoms were evaluated according to the criteria for long-COVID, identifying those who experienced long-term health effects after the infection. At the time of follow-up, 57 patients completed the questionnaire; 29 were classified as long-COVID due to persistent multi-system symptoms, while the rest had no such symptoms. Collecting this data enabled a comprehensive assessment of the long-term consequences of COVID-19 and correlation with earlier immunological findings. Long-COVID was defined according to the World Health Organization (WHO) criteria, as symptoms starting at least three months after infection and lasting for a minimum of two months (https://www.who.int/europe/news-room/fact-sheets/item/post-covid-19-condition, 30 July 2025).

### 2.6. Statistical Elaboration

Statistical analysis was conducted using Statistica 13.3 software (TIBCO Software Inc., Palo Alto, CA, USA) and R (version 3.6.1). Categorical variables were presented as numbers (percentages), and group differences were calculated using the Chi^2^ or exact Fisher test, as appropriate. The distribution of continuous data was assessed using the Shapiro–Wilk test. All data were non-normally distributed and thus are presented as medians with Q1–Q3 ranges and compared using the Mann–Whitney test. To adjust for covariates, non-normally distributed data were Box-Cox transformed, and a one-way analysis of covariance (ANCOVA) was performed with adjustments for age, sex, and body mass index. Missing data were handled by calculating percentages based on the available data for each variable. Results with *p*-values below 0.05 were recorded as statistically significant.

## 3. Results

### 3.1. Characteristics of the Patients

Our study comprised 103 COVID-19 patients, 53 with severe COVID-19 (51.5%) and 50 with mild disease (48.5%). Detailed information, including demographic, clinical, and laboratory characteristics of our cohort, is provided in Table 1.

As presented, both groups were similar in terms of age, sex, and body mass index (BMI). Regarding clinical characteristics, the most common concomitant disease was arterial hypertension, followed by obesity and hypercholesterolemia, regardless of group affiliation. Interestingly, patients with a mild form of an acute disease presented more often with a history of recurrent infections, also after adjustment for potential confounders (Table 1).

According to clinical symptoms during hospitalization, the severe COVID-19 group suffered more often from fever, but less frequently from pleuritic pain, than patients with mild disease. Moreover, as expected, in severe disease, we observed lower oxygen saturation of peripheral blood and an increased number of points on the PESI scale at admission to the hospital (*p* < 0.05 for all mentioned variables, adjusted for age, sex, and BMI). Regarding therapy, severe COVID-19 patients were more frequently treated with intermediate and therapeutic low molecular weight heparin (LMWH) and with antibiotics during hospitalization. For additional details, refer to Table 1.

### 3.2. Laboratory Parameters in Severe and Mild COVID-19 Patients

Data on blood laboratory test results are provided in Table 2. Severe COVID-19 patients had elevated leukocyte counts, with higher neutrophil and lower lymphocyte counts. Next, we observed an increase in D-dimer levels in the severe COVID-19 group, along with troponin and NT-proBNP concentrations. Furthermore, this group was characterized by worse kidney function, higher glucose and uric acid, and increased activity of liver enzymes. As expected, patients with severe disease also exhibited more pronounced inflammatory parameters, including CRP, procalcitonin, and IL-6. There were no differences in natural anticoagulant proteins between groups; however, severe COVID-19 patients had elevated concentrations of anticardiolipin antibodies in both classes and anti-beta-2 glycoprotein I antibodies in the IgM class, even after adjustment for age, sex, and BMI (Table 2).

### 3.3. Immunological Parameters in Severe and Mild COVID-19 Patients at Baseline

In terms of circulating lymphocyte subsets, patients with severe COVID-19 presented a tendency toward a decrease in certain lymphocyte subpopulations (Table 3). Specifically, they were characterized by lower CD3+ T cells (count and percentage), CD4+ T cells (count and percentage), CD8+ T cells (count), and NK cells (count). In turn, we did not notice a difference in the CD4+/CD8+ ratio between severe and mild COVID-19 subgroups. Regarding B lymphocytes, although there was no significant difference in the absolute counts of CD19+ B lymphocytes between the two groups, the percentage of these cells was significantly higher in patients with severe COVID-19 compared to those with mild disease (*p* = 0.002). We also noticed differences in ratios between selected subtypes of lymphocytes (Table 3). Despite these variations in lymphocyte subsets, no differences were observed in the levels of total IgG or IgE between the two groups (Table 3).

### 3.4. Follow-Up Analysis with a Focus on Baseline Immune-Related Variables and Later Death or Presence of Long-COVID Symptoms

To investigate whether immune alterations in the acute phase of COVID-19 were linked to long-term outcomes, we analyzed late disease outcomes at the time of follow-up. Among the 103 patients initially enrolled, data on later outcomes were available for 80 individuals (77.7%), while 23 patients (22.3%) were lost to follow-up due to lack of contact, refusal to participate, or withdrawal of consent. Of the 80 patients with available follow-up data, 23 (28.8%) had died, while 57 (71.2%) were confirmed alive at the median 5-year follow-up. We explored potential associations between demographic, clinical, and laboratory characteristics, including immune parameters during the acute COVID-19 phase and long-term outcomes, such as death or presence of long-COVID-19 symptoms. A summary of clinical baseline characteristics is presented in Table 4. In the deceased group, patients were older and had a more severe disease course; however, there were no significant differences in the frequency of comorbidities compared to survivors. Interestingly, the duration of hospitalization was shorter in the deceased group.

Then we examined whether baseline immune dysregulation, including lymphocyte subpopulations and immunoglobulin levels, differed between those who later died and those who survived (Table 5). Patients who died during follow-up had significantly lower absolute counts of CD3+ T cells, CD4+ T cells, CD8+ T cells, and NK cells compared to survivors of COVID-19. Notably, the percentage of CD19+ B lymphocytes was also higher in deceased patients (*p* = 0.049), whereas other parameters, such as the CD4+/CD8+ ratio, immunoglobulin levels, and percentages of CD3+, CD4+, and CD8+ cells, showed no significant differences. Consequently, we also noticed differences in ratios between selected subtypes of lymphocytes (Table 5).

Next, among the 80 patients in the follow-up group, we focused on those who had a severe initial course of COVID-19 (*n* = 40). We then compared clinical and laboratory data between survivors and those who died in this subgroup, aiming to identify potential predictors of survival in severe COVID-19 (Table 6). Patients who died were older with a more severe disease course (lower saturation and higher PESI index) but were hospitalized for a shorter period compared to survivors. Regarding the lymphocytes, no significant differences were observed (Table 7).

### 3.5. Baseline Differences in Immunological Parameters in Patients with and Without Long-COVID Symptoms in the Follow-Up

Finally, among the surviving patients, we analyzed the associations between baseline demographic, clinical, and laboratory features, including immune dysregulation, and the development of long-COVID symptoms. Among 57 surviving patients, about half reported symptoms related to long-COVID (*n* = 29, 50.9%). Patients most frequently report cognitive impairment (including memory and concentration problems), persistent fatigue, dyspnea, joint pain, and sensory disturbances, including loss of smell, taste, or hearing. A summary of baseline characteristics of these patients is provided in Table 8. There were no significant clinical differences between the analyzed subgroups, except for a lower frequency of smell and taste disorders in the acute COVID-19 phase in patients with long-COVID symptoms.

In the follow-up, patients with long-COVID reported significantly longer recovery times from the acute disease phase, with only 17.2% recovering within three months compared to 53.6% in the group without long-COVID (*p*<0.001) (Table 9). Most long-COVID patients (55.2%) required more than six months to recover, in contrast to only 10.7% in the non-long-COVID subgroup. Among patients diagnosed with long-COVID, 25 individuals (86.2%) reported at least one of the following: chronic fatigue, dyspnoea, cognitive impairment (“brain fog”), persistent loss of smell or taste, chest pain, palpitations, cough, myalgia, sleep disturbances, and dizziness. Persistent headaches, thromboembolic events, newly diagnosed cardiac, neurological, or renal conditions, as well as further episodes of COVID-19 and vaccination status, were comparable between the groups (Table 9).

Surprisingly, there were no differences in the baseline immunological parameters between patients with and without long-COVID symptoms in the follow-up analysis (Table 10).

## 4. Discussion

In this study, we provided a comparative analysis of immunological, clinical, and laboratory characteristics in patients with severe and mild COVID-19, and further explored the long-term outcomes, including mortality rate and long-COVID symptoms, at follow-up. The study supported the notion that early immune dysregulation predicts disease severity and long-term prognosis, particularly in association with later mortality. We did not find links with long-COVID symptoms; however, heterogeneity of signs, which are encompassed under a single term of “long-COVID”, may have obscured more specific associations.

In the acute COVID-19 phase, we observed differences between severe and mild cases, including significantly lower absolute counts of CD3+ T lymphocytes, CD4+ helper T cells, CD8+ cytotoxic T cells, and NK cells in the severe subgroup. Thus, our results indicate an impaired cellular immunity in the early stage of severe COVID-19. These alterations were accompanied by elevated markers of systemic inflammation, such as CRP, PCT, IL-6, and LDH, and higher levels of organ dysfunction markers, including ALT, AST, urea, D-dimer, and troponin. Therefore, our results are consistent with previous studies indicating that decreased lymphocyte count and elevated inflammatory markers are hallmarks of severe COVID-19 and may reflect immune exhaustion and cytokine storm phenomena [16,17]. For example, Zhao et al. [18] revealed that lymphopenia, lower CD4+, and CD3+ T cell counts were immunity-related risk factors predicting mortality in patients with acute COVID-19. Likewise, Andrejkovits et al. [19] found that the absolute number of CD3+ lymphocytes might independently predict the severity of COVID-19 and fatal outcomes in COVID-19 patients. Another study by Gygi et al. [20] demonstrates that not only lymphopenia but also dysregulated T cell functions, including aberrant T cell apoptosis, contribute to increased COVID-19 severity and risk of death.

We did not analyze whether the immune dysregulation observed in the acute COVID-19 phase remains during the follow-up; however, a study by Putri et al. [21] revealed lower total T and CD4+ T cell counts in the long-COVID-19 group (mild and severe subgroups), with increased CD8+ T cells compared to the group without COVID-19 infection and vaccination. Krishna et al. [22] revealed that persistent, antigen-independent IFN-γ release by CD8^+^ T cells, dependent on CD14^+^ cells, may represent an underlying mechanism of long-COVID and a potential biomarker for disease persistence and recovery. In a study by Vijayakumar et al. [23], long-COVID was associated with persistent immunological and proteomic alterations in the airways, characterized by cytotoxic T cell activity, epithelial injury, and tissue repair processes, which gradually resolved over time. Interestingly, Klein et al. [24] presented evidence that long-COVID was also characterized by distinct immune dysregulation, including altered myeloid and lymphocyte populations, and exaggerated antiviral responses, but on the other hand also by a reduced cortisol level, highlighting potential biomarkers and pathways underlying persistent symptoms. All these observations suggest that immunological changes persist long after the recovery.

In the follow-up analysis, 80 patients were included out of 103 at the study baseline. Among these patients with available data, those who died within the observation period had significantly lower initial counts of CD3+, CD4+, CD8+, and NK cells compared to survivors. This suggests that early immune-related disturbances may contribute to the acute severity of the condition and may also be associated with poorer long-term outcomes. For instance, a study by Balzanelli et al. [25] on COVID-19 patients revealed that alterations in peripheral lymphocyte subsets were associated with the clinical characteristics and progression of COVID-19. In the context of long-COVID, data suggest that both humoral and cellular immune responses play crucial roles as potential predictors of long-term complications [24,26]. The humoral response, characterized by the overproduction of antibodies such as IgA, IgG, and IgM, has been linked to protective effects. For instance, as reported in a study by Cervia et al. [27], an early and robust IgA/IgG response appears to correlate with efficient viral clearance and a reduced risk of developing long-COVID symptoms. In contrast, dysregulation in the cellular immune response, characterized by elevated levels of pro-inflammatory cytokines (including IL-1β, IL-6, and TNF), altered T cell (CD4+ and CD8+) profiles, and monocyte activation, has been frequently linked to persistent inflammation and chronic symptoms, including neuropsychiatric disturbances [28,29,30]. Moreover, a balanced immune profile may be critical for predicting patients’ death, as observed by Ruytinx et al. [31]. In this study, higher baseline levels of IL-8, IL-6, IL-10, IFN-α, IFN-β, IFN-γ, and IFN-λ1 were associated with increased mortality. In contrast, increased circulating concentrations of IFN-λ2 and IFN-λ3 played a protective role in a cohort of Belgian COVID-19 patients.

Long-COVID-19 is believed to be related to chronic inflammation and immune dysregulation, including hyperactivity of innate immune cells, and elevated pro-inflammatory cytokines [32]. In our cohort, long-COVID symptoms were prevalent in more than half of the survivors, with cognitive dysfunction, fatigue, and dyspnea among the most reported complaints. However, in contrast to mortality risk, no significant differences were observed in baseline immune profiles between patients who developed long-COVID symptoms and those who did not. This finding may suggest that long-COVID is driven more by post-infectious dysregulation, metabolic or autonomic disturbances, or ongoing inflammation, rather than by initial quantitative immune suppression as illustrated in the current literature [33,34]. The absence of significant differences in baseline T and B cell counts, immunoglobulin levels, or NK cell counts between long-COVID and non-long-COVID patients also suggests the likely multifactorial nature of this syndrome, involving psychosocial, neurological, and microvascular components that traditional immunological assays may not capture. In this context, long-COVID appears to be more related to ongoing immune dysregulation and chronic inflammation rather than the initial severity of lymphocyte depletion [35]. However, some literature data contradict our findings. For example, in the study by Phetsouphanh et al. [36], patients with long-COVID had highly activated innate immune cells, lacked naive T and B cells, and exhibited elevated expression of IFN-β and IFN-λ1 at 8 months post-COVID-19. In turn, LaVergne et al. [37] have demonstrated that in the acute COVID-19 phase, CD8+ (CD8+ Ki67+) T cells were significantly higher in patients who developed persistent dyspnea, while CD4+ CD25+ T cells were increased in those who later developed persistent forgetfulness. Therefore, the degree of activation of individual T lymphocytes is likely important in the development of different long-COVID-19 characteristics.

Interestingly, in our follow-up cohort, we did not observe differences in vaccination status between patients with and without long-COVID. However, our study was conducted on patients who suffered from COVID-19 between June and November 2020, that is, before the vaccines were available. That is likely why vaccination status has no impact on the frequency of long-COVID symptoms in our data. There is some evidence that vaccination before the disease onset decreases the risk of long-COVID development but has no impact on those with established long-COVID [38], or may alleviate certain long-COVID manifestations, such as severe fatigue [39] or others [40,41], but not cardiovascular complications [42].

Although the CD4+/CD8+ ratio and total immunoglobulin levels did not differ significantly between survivors and non-survivors, the higher percentage of CD19+ B lymphocytes in the deceased group raises questions about the functional quality of humoral responses in these patients. In fact, the severity of COVID-19 may be associated with changes in B cell subpopulations, both immature and terminally differentiated [43]. Next, Mendez-Cortina et al. [44] observed a significant increase in CD19+ B cells in hospitalized and intensive care unit patients compared to asymptomatic and symptomatic groups. On the other hand, Çölkesen et al. [45] reported lower numbers of different B cell subsets in deceased patients, including total, naive, and switched memory B cells. Therefore, these reports suggest that both the quantity and functionality of B cells are important for COVID-19 outcomes.

Patients with severe disease also had significantly increased levels of antiphospholipid antibodies, which may suggest an autoimmune component in the pathogenesis of severe SARS-CoV-2 infection. That aligns with earlier reports suggesting that antiphospholipid antibodies may contribute to the hypercoagulable state observed in COVID-19 [46], although their direct pathogenic role remains controversial [47]. Clinical symptomatology also differed between the two severity groups. Severe patients presented more frequently with fever and lower oxygen saturation, as well as elevated Pulmonary Embolism Severity Index (PESI) scores. Treatment approaches also diverged significantly: severe patients were more frequently administered intermediate or therapeutic doses of low molecular weight heparin, in contrast to the predominance of prophylactic LMWH in mild cases. These results are also in line with other reports on COVID-19 [48,49].

This study has several limitations. Firstly, the sample size, particularly in the long-term follow-up, was relatively small, with only 80 of the original 103 patients available. Secondly, the study was conducted at a single center, which may limit the generalizability of the findings to broader populations with different demographics, healthcare systems, or viral variants. Thirdly, although detailed immunological profiling was performed at baseline, the lack of serial measurements over time prevents assessment of the dynamic changes in immune parameters. Additionally, we analyzed differences but did not assess the strength of the associations. Next, the evaluation of long-COVID relied on retrospective self-reported symptoms, with the possibility of recall bias, including detailed information on prior COVID-19 vaccination history and the frequency of infections in follow-up. Finally, while we explored associations between immune variables and long-term outcomes, causality cannot be inferred from this observational design. Confounding factors such as comorbidities, treatments received, or psychosocial determinants may also have influenced the results. However, we believe that every report on understanding long-COVID is valuable, as many patients continue to struggle with it, and more research is urgently needed to clarify its underlying mechanisms, risk factors, and potential therapeutic targets.

## 5. Conclusions

Lymphopenia, especially in T lymphocyte subsets and NK cells, was associated with both acute disease severity and long-term mortality. Baseline immune parameters were not predictive of long-COVID, suggesting potentially different underlying mechanisms. These findings suggest that long-COVID is likely driven by complex, multifactorial mechanisms beyond initial immune suppression, underscoring the need for integrated long-term patient monitoring and personalized approaches to care.

## Figures and Tables

**Table 1 viruses-17-01400-t001:** Demographic and clinical characteristics of COVID-19 patients.

Parameter	Severe COVID-19 Patients *n* = 53	Mild COVID-19 Patients *n* = 50	*p*-Value (Adjusted for Age, Sex, and BMI)
**Demographic characteristics**			
Age, years	63.0 (53.0–71.0)	59.5 (49.0–67.0)	0.14
Sex, male, *n* (%)	41 (77.4%)	30 (60.0%)	0.06
Body mass index, kg/m^2^	29.8 (26.0–33.0)	28.4 (25.8–33.3)	0.46
**Comorbidities**			
Hypertension, *n* (%)	33 (62.3%)	25 (50.0%)	0.21
Diabetes mellitus, *n* (%)	14 (26.4%)	9 (18.0%)	0.31
Hypercholesterolemia, *n* (%)	19 (35.9%)	12 (24.0%)	0.19
Obesity, *n* (%)	21 (39.6%)	14 (28.0%)	0.21
Atrial fibrillation, *n* (%)	4 (7.6%)	4 (8.0%)	0.93
Chronic heart failure, *n* (%)	11 (20.8%)	6 (12.0%)	0.23
Coronary artery disease, *n* (%)	11 (20.8%)	9 (18.0%)	0.72
Peripheral artery obliterans disease, *n* (%)	5 (9.4%)	3 (6.0%)	0.72
eGFR < 50 mL/min/1.73 m^2^, *n* (%)	7 (13.2%)	2 (4.0%)	0.16
Asthma, *n* (%)	5 (9.4%)	7 (14.0%)	0.47
Malignancy, *n* (%)	3 (5.7%)	2 (4.0%)	>0.99
Autoimmune diseases, *n* (%)	5 (9.4%)	6 (12.0%)	0.67
History of recurrent infections, *n* (%)	0 (0.0%)	7 (14.0%)	**0.005**
**Clinical symptoms in the acute COVID-19 phase**			
Fever, *n* (%)	46 (86.8%)	34 (68.0%)	**0.022**
Dyspnea, *n* (%)	47 (88.7%)	38 (78.0%)	0.09
Pleuritic pain, *n* (%)	2 (3.8%)	9 (18.0%)	**0.026**
Cough, *n* (%)	44 (83.0%)	34 (68.0%)	0.08
Smell and taste disorders, *n* (%)	11 (20.8%)	16 (32.0%)	0.20
Muscle pain, *n* (%)	10 (18.9%)	17 (34.0%)	0.08
**Parameters at admission to the hospital**			
Systolic blood pressure, mmHg	130 (120–138)	130 (120–139)	0.96
Diastolic blood pressure, mmHg	79 (70–85)	80 (70–89)	0.28
Saturation, %	92 (86–95)	96 (94–97)	**0.003**
Temperature, °C	36.6 (36.6–36.6)	36.6 (36.4–36.8)	0.30
Heart rate, beats per minute	83 (72–93)	82 (75–95)	0.91
Wells Score, points	0 (0–0)	0 (0–0)	0.82
Geneva Score, points	3 (1–5)	3 (0–4)	0.39
Pulmonary Embolism Severity Index, points	99 (83–116)	90 (75–99)	**0.003**
Pack-years, number	0 (0–10)	0 (0–0)	0.47
**Treatment pattern before and after admission to the hospital**			
Aspirin, *n* (%)	13 (24.5%)	6 (12.0%)	0.17
Direct oral anticoagulants, *n* (%)	5 (9.4%)	3 (6.0%)	0.78
Beta blockers, *n* (%)	27 (50.9%)	17 (34.0%)	0.12
Angiotensin II receptor blockers or sartans, *n* (%)	11 (20.8%)	9 (18.0%)	0.92
Statins, *n* (%)	19 (35.9%)	13 (26.0%)	0.39
Diuretics, *n* (%)	15 (28.3%)	15 (30.0%)	0.98
Glucocorticosteroids, orally, *n* (%)	4 (7.6%)	8 (16.0%)	0.30
Glucocorticosteroids, inhaled, *n* (%)	5 (9.4%)	7 (14.0%)	0.68
Long-acting beta 2 agonist, *n* (%)	5 (9.4%)	7 (14.0%)	0.68
Prophylactic LMWH during hospitalization, *n* (%)	6 (11.3%)	25 (50.0%)	**<0.001**
Intermediate LMWH during hospitalization, *n* (%)	32 (60.4%)	17 (34.0%)	**0.013**
Therapeutic LMWH, *n* (%)	32 (60.4%)	13 (26.0%)	**<0.001**
Antibiotics during hospitalization, *n* (%)	53 (100.0%)	43 (86.0%)	**0.015**
**Number of hospitalization days**			
Number of hospitalization days	15 (10–23)	13 (8–20)	0.33

Categorical variables are presented as numbers with percentages. Continuous variables as medians and Q1–Q3 ranges. Statistically significant differences are in bold. Abbreviations: *n*—number; eGFR—estimated glomerular filtration rate; LMWH—low molecular weight heparin, BMI—body mass index.

**Table 2 viruses-17-01400-t002:** Baseline laboratory characteristics of COVID-19 patients.

Parameter	Severe COVID-19 Patients *n* = 53	Mild COVID-19 Patients *n* = 50	*p*-Value (Adjusted for Age, Sex, and BMI)
**Laboratory characteristics**			
White blood cells, 10^3^/µL	9.2 (6.7–12.6)	6.5 (5.0–8.9)	**<0.001**
Neutrophils, 10^3^/µL	7.5 (5.9–9.8)	4.7 (3.3–6.9)	**0.028**
Lymphocytes, 10^3^/µL	0.8 (0.5–1.1)	1.1 (0.8–1.4)	**0.040**
Eosinophils, 10^3^/µL	0.02 (0.00–0.22)	0.02 (0.00–0.12)	0.58
Basophils, 10^3^/µL	0.03 (0.02–0.04)	0.03 (0.02–0.05)	0.46
Red blood cells, 10^6^/µL	4.3 (3.7–4.5)	4.4 (4.1–4.8)	0.74
Hemoglobin, g/dL	13.0 (12.3–14.4)	13.4 (12.5–14.2)	0.53
Blood platelets, 10^3^/µL	220 (167–275)	213 (163–248)	0.97
D-dimer, mg/mL	1.00 (0.71–5.07)	0.67 (0.43–1.12)	**<0.001**
Troponin I, ng/mL	8.14 (4.04–91.65)	5.56 (2.54–11.97)	**0.028**
Fibrinogen, g/L	5.60 (4.80–6.51)	5.70 (4.44–6.20)	0.75
Activated partial thromboplastin time, s	33.9 (30.9–37.7)	30.5 (28.0–34.6)	**0.011**
International normalized ratio	1.0 (0.9–1.1)	1.0 (0.9–1.1)	0.95
Creatinine, µmol/L	79 (65–111)	77 (64–91)	0.49
Urea, mmol/L	7.5 (5.9–11.7)	4.9 (4.2–7.0)	**0.018**
Alanine transaminase, U/L	48.0 (32.0–74.0)	30.5 (22.5–50.5)	**0.019**
Aspartate transaminase, U/L	62.0 (44.0–85.0)	35.5 (25.0–47.0)	**<0.001**
Glucose, mmol/L	8.0 (6.1–9.9)	6.2 (5.3–7.7)	**0.036**
Bilirubin, μmol/L	9.8 (6.1–13.7)	6.6 (4.8–10.9)	**0.042**
Gamma-glutamyl transpeptidase, U/L	57.5 (28.5–133.0)	46.0 (25.0–74.0)	0.08
Lactate dehydrogenase, IU/L	529 (456–615)	311 (228–421)	**<0.001**
Creatine kinase, U/L	132 (68–378)	97 (66–201)	0.50
Myoglobin, ng/mL	93.3 (60.9–160.4)	66.9 (41.1–167.7)	0.46
NT-proBNP, pg/mL	604 (242–2423)	239 (133–1346)	**0.024**
C-reactive protein, mg/L	143.0 (84.0–228.0)	50.5 (19.9–116.0)	**0.015**
Procalcitonin, ng/mL	0.175 (0.060–0.345)	0.040 (0.020–0.070)	**<0.001**
Interleukin-6, pg/mL	66.5 (31.0–153.1)	35.9 (15.5–73.7)	**0.015**
C3c, g/L	1.52 (1.28–1.81)	1.50 (1.31–1.79)	0.64
C4, g/L	0.31 (0.25–0.36)	0.35 (0.28–0.40)	0.05
**Anticoagulant proteins**			
Activity of antithrombin, %	96.4 (89.9–109.7)	102.1 (94.1–109.4)	0.40
Free protein S level, %	79.3 (67.0–91.8)	82.8 (67.9–100.6)	0.16
Activity of protein C, %	107.1 (90.1–137.9)	103.2 (89.7–115.8)	0.44
Activated protein C resistance, ratio	3.10 (2.60–3.40)	2.85 (2.70–3.20)	0.45
**Antiphospholipid antibodies**			
Anticardiolipin antibodies IgM, MPL	15.88 (10.86–28.47)	10.88 (8.10–13.38)	**0.002**
Anticardiolipin antibodies IgG, GPL	7.58 (4.16–14.49)	4.66 (3.49–6.88)	**0.008**
Anti-β2 glycoprotein I antibodies IgM, SMU	2.82 (1.63–5.07)	1.80 (1.40–3.22)	**0.025**
Anti-β2 glycoprotein I antibodies IgG, SGU	1.70 (1.36–2.42)	1.79 (1.28–2.27)	0.98

Continuous variables are presented as medians and Q1–Q3 ranges. Statistically significant differences are in bold. Abbreviations: *n*—number, BMI—body mass index.

**Table 3 viruses-17-01400-t003:** Detailed immunological laboratory characteristics of COVID-19 patients.

Parameter	Severe COVID-19 Patients *n* = 53	Mild COVID-19 Patients *n* = 50	*p*-Value (Adjusted for Age, Sex, and BMI)
**Immunological characteristics**			
Lymphocytes CD3+, *n*/µL	340 (213–544)	705 (586–1022)	**0.016**
Lymphocytes CD3+, %	56.5 (48.1–62.8)	66.5 (57.4–74.7)	**0.025**
Helper lymphocytes CD4+, *n*/µL	183 (114–360)	452 (301–595)	**0.002**
Lymphocytes helper CD4+, %	31.4 (24.3–38.6)	39.0 (29.8–44.5)	**0.003**
Cytotoxic lymphocytes CD8+, *n*/µL	109 (69–165)	227 (158–317)	**<0.001**
Lymphocytes cytotoxic CD8+, %	15.6 (10.5–24.3)	13.1 (9.1–22.4)	0.92
Lymphocytes NK, *n*/µL	107 (54–161)	157 (119–241)	**0.004**
CD4+/CD8+ ratio	1.93 (1.32–2.88)	2.14 (1.45–2.59)	0.71
Lymphocytes B CD19+, *n*/µL	117 (70–219)	136 (100–199)	0.86
Lymphocytes B CD19+, %	18.5 (11.7–27.6)	12.5 (8.7–16.9)	**0.044**
Immunoglobulin G, g/L	10.4 (9.2–11.8)	11.3 (9.4–12.9)	0.18
Immunoglobulin E, IU/mL	54.9 (24.0–211.0)	81.3 (29.3–247.0)	0.39
**Detailed ratio characteristics**			
Ratio: Lymphocytes B CD19+, %/Lymphocytes helper CD4+, %	0.16 (0.10–0.22)	0.09 (0.07–0.12)	**<0.001**
Ratio: Lymphocytes B CD19+, %/Lymphocytes cytotoxic CD8+, %	1.23 (0.74–2.15)	0.92 (0.64–1.99)	0.14
Ratio: Lymphocytes B CD19+, %/Lymphocytes CD3+, %	0.35 (0.23–0.55)	0.19 (0.13–0.30)	**<0.001**
Ratio: Lymphocytes B CD19+, *n*/µL/Lymphocytes helper CD4+, *n*/µL	0.56 (0.42–0.98)	0.34 (0.23–0.54)	**<0.001**
Ratio: Lymphocytes B CD19+, *n*/µL/Lymphocytes cytotoxic CD8+, *n*/µL	1.18 (0.66–2.15)	0.58 (0.42–1.12)	**<0.001**
Ratio: Lymphocytes B CD19+, *n*/µL/Lymphocytes CD3+, *n*/µL	0.35 (0.23–0.55)	0.19 (0.13–0.30)	**<0.001**
Ratio: Lymphocytes B CD19+, *n*/µL/Lymphocytes NK, *n*/µL	1.37 (0.68–2.07)	0.92 (0.63–1.43)	0.07

Continuous variables are presented as medians and Q1–Q3 ranges. Statistically significant differences are in bold. Abbreviations: *n*—number, NK—natural killer, CD—cluster of differentiation, BMI—body mass index.

**Table 4 viruses-17-01400-t004:** Characteristics of COVID-19 patients from the follow-up regarding baseline parameters.

Parameter	Deceased COVID-19 Patients in the Follow-Up *n* = 23	Surviving COVID-19 Patients in the Follow-Up *n* = 57	*p*-Value
**Demographic characteristics**			
Age, years	66.0 (58.5-74.5)	56.0 (47.0-66.0)	**0.004**
Sex, male, *n* (%)	18 (78.3%)	37 (64.9%)	0.24
Body mass index, kg/m^2^	28.7 (26.4-32.9)	30.0 (26.1-33.5)	0.67
**Comorbidities**			
Hypertension, *n* (%)	12 (52.2%)	33 (57.9%)	0.64
Diabetes mellitus, *n* (%)	5 (21.7%)	11 (19.3%)	0.80
Hypercholesterolemia, *n* (%)	9 (39.1%)	16 (28.1%)	0.33
Obesity, *n* (%)	8 (34.8%)	21 (36.8%)	0.86
Atrial fibrillation, *n* (%)	3 (13.0%)	3 (5.3%)	0.35
Chronic heart failure, *n* (%)	5 (21.7%)	6 (10.5%)	0.19
Coronary artery disease, *n* (%)	6 (26.1%)	7 (12.8%)	0.13
Peripheral artery obliterans disease, *n* (%)	3 (13.0%)	3 (5.3%)	0.35
eGFR < 50 mL/min/1.73 m^2^, *n* (%)	4 (17.4%)	2 (3.5%)	0.05
Asthma, *n* (%)	1 (4.3%)	11 (19.3%)	0.16
Malignancy, *n* (%)	3 (13.0%)	2 (3.5%)	0.14
Autoimmune diseases, *n* (%)	3 (13.0%)	7 (12.3%)	>0.99
History of recurrent infections, *n* (%)	0 (0.0%)	5 (8.8%)	0.31
**Clinical symptoms in the acute COVID-19 phase**			
Fever, *n* (%)	20 (87.0%)	44 (77.2%)	0.32
Dyspnea, *n* (%)	19 (82.6%)	45 (78.9%)	0.71
Pleuritic pain, *n* (%)	1 (4.3%)	8 (14.0%)	0.43
Cough, *n* (%)	15 (65.2%)	44 (77.2%)	0.27
Smell and taste disorders, *n* (%)	3 (13.0%)	19 (33.3%)	0.10
Muscle pain, *n* (%)	3 (13.0%)	17 (29.8%)	0.16
**Parameters at admission to the hospital**			
Systolic blood pressure, mmHg	130 (122–140)	130 (120–136)	0.41
Diastolic blood pressure, mmHg	76 (70–86)	80 (70–85)	0.59
Saturation, %	91.0 (88.5–94.0)	95.5 (94.0–96.3)	**<0.001**
Temperature, °C	36.6 (36.6–36.6)	36.6 (36.5–36.7)	0.57
Heart rate, beats per minute	80 (70–106)	81 (72–91)	0.89
Wells Score, points	0 (0–1)	0 (0–0)	0.11
Geneva Score, points	4 (1–5)	3 (0–4)	**0.04**
Pulmonary Embolism Severity Index, points	103 (91–123)	86 (75–97)	**<0.001**
Pack-years, number	0 (0–15)	0 (0–1)	0.29
**Treatment pattern before and after admission to the hospital**			
Aspirin, *n* (%)	9 (39.1%)	7 (12.3%)	**0.007**
Direct oral anticoagulants, *n* (%)	3 (13.0%)	2 (3.5%)	0.14
Beta blockers, *n* (%)	10 (43.5%)	22 (38.6%)	0.69
Angiotensin II receptor blockers/sartans, *n* (%)	4 (17.4%)	13 (22.8%)	0.77
Statins, *n* (%)	8 (34.8%)	17 (29.8%)	0.67
Diuretics, *n* (%)	4 (17.4%)	18 (31.6%)	0.27
Glucocorticosteroids, orally, *n* (%)	2 (8.7%)	7 (12.3%)	>0.99
Glucocorticosteroids, inhaled, *n* (%)	2 (8.7%)	10 (17.5%)	0.49
Long-acting beta 2 agonist, *n* (%)	2 (8.7%)	10 (17.5%)	0.49
Prophylactic LMWH during hospitalization, *n* (%)	5 (21.7%)	21 (36.8%)	0.19
Intermediate LMWH during hospitalization, *n* (%)	13 (56.5%)	28 (49.1%)	0.55
Therapeutic LMWH, *n* (%)	14 (60.9%)	17 (29.8%)	**0.010**
Antibiotics during hospitalization, *n* (%)	22 (95.7%)	51 (89.5%)	0.38
**Details on hospitalization**			
Number of hospitalization days	12 (3–17)	17 (11–22)	**0.011**
Severe course of COVID-19, *n* (%)	20 (87.0%)	20 (35.1%)	**<0.001**

Categorical variables are presented as numbers with percentages. Continuous variables as medians and Q1–Q3 ranges. Statistically significant differences are in bold. Abbreviations: *n*—number; eGFR—estimated glomerular filtration rate; LMWH—low molecular weight heparin.

**Table 5 viruses-17-01400-t005:** Comparison of baseline immunological parameters between deceased and surviving COVID-19 patients at 5-year follow-up.

Parameter	Deceased COVID-19 Patients in the Follow-Up *n* = 23	Surviving COVID-19 Patients in the Follow-Up *n* = 57	*p*-Value
**Immunological characteristics**			
Lymphocytes CD3+, *n*/µL	377 (246–595)	655 (395–972)	**0.005**
Lymphocytes CD3+, %	60.0 (50.1–64.9)	63.6 (54.3–72.6)	0.16
Lymphocytes helper CD4+, *n*/µL	224 (110–404)	372 (210–490)	**0.021**
Lymphocytes helper CD4+, %	32.0 (25.4–42.7)	36.3 (29.1–42.1)	0.48
Lymphocytes cytotoxic CD8+, *n*/µL	113 (58–152)	188 (134–273)	**<0.001**
Lymphocytes cytotoxic CD8+, %	17.3 (13.1–23.2)	14.2 (9.9–24.3)	0.46
Lymphocytes NK, *n*/µL	118 (54–151)	157 (99–236)	**0.010**
CD4+/CD8+ ratio	2.19 (1.21–3.29)	1.86 (1.40–2.37)	0.35
Lymphocytes B CD19+, *n*/µL	128 (68–229)	127 (83–191)	0.75
Lymphocytes B CD19+, %	19.2 (11.4–25.9)	13.5 (9.6–17.0)	**0.049**
Immunoglobulin G, g/L	10.4 (9.7–10.8)	11.1 (9.2–13.0)	0.19
Immunoglobulin E, IU/mL	150.0 (38.0–352.0)	68.5 (27.0–232.0)	0.47
**Detailed ratio characteristics**			
Ratio: Lymphocytes B CD19+, %/Lymphocytes helper CD4+, %	0.14 (0.09–0.25)	0.10 (0.07–0.14)	**0.019**
Ratio: Lymphocytes B CD19+, %/Lymphocytes cytotoxic CD8+, %	1.09 (0.61–1.45)	0.89 (0.49–1.39)	0.43
Ratio: Lymphocytes B CD19+, %/Lymphocytes CD3+, %	0.29 (0.19–0.51)	0.23 (0.14–0.31)	0.05
Ratio: Lymphocytes B CD19+, *n*/µL/Lymphocytes helper CD4+, *n*/µL	0.57 (0.34–0.87)	0.41 (0.25–0.55)	**0.025**
Ratio: Lymphocytes B CD19+, *n*/µL/Lymphocytes cytotoxic CD8+, *n*/µL	1.12 (0.58–2.15)	0.66 (0.44–1.24)	**0.024**
Ratio: Lymphocytes B CD19+, *n*/µL/Lymphocytes CD3+, *n*/µL	0.29 (0.19–0.51)	0.23 (0.14–0.31)	**0.048**
Ratio: Lymphocytes B CD19+, *n*/µL/Lymphocytes NK, *n*/µL	1.21 (0.66–1.71)	0.89 (0.45–1.37)	0.18

Continuous variables are presented as medians with Q1–Q3 ranges. Statistically significant differences are in bold. Abbreviations: *n*—number, NK—natural killer, CD—cluster of differentiation.

**Table 6 viruses-17-01400-t006:** Baseline clinical and laboratory characteristics of severe COVID-19 patients stratified by survival status at follow-up.

Parameter	Deceased COVID-19 Patients in the Follow-Up Among Baseline Severe COVID-19 Cases *n* = 20	Surviving COVID-19 Patients in the Follow-Up Among Baseline Severe COVID-19 Cases *n* = 20	*p*-Value
**Demographic characteristics**			
Age, years	67.0 (59.5–77.0)	53.5 (49.0–66.5)	**0.007**
Sex, male, *n* (%)	16 (80.0%)	15 (75.0%)	>0.99
Body mass index, kg/m^2^	28.7 (26.7–32.9)	30.5 (27.4–33.3)	0.53
**Comorbidities**			
Hypertension, *n* (%)	12 (60.0%)	14 (70.0%)	0.51
Diabetes mellitus, *n* (%)	5 (25.0%)	5 (25.0%)	>0.99
Hypercholesterolemia, *n* (%)	8 (40.0%)	8 (40.0%)	>0.99
Obesity, *n* (%)	7 (35.0%)	10 (50.0%)	0.34
Atrial fibrillation, *n* (%)	3 (15.0%)	0 (0.0%)	0.23
Chronic heart failure, *n* (%)	5 (25.0%)	2 (10.0%)	0.41
Coronary artery disease, *n* (%)	6 (30.0%)	2 (10.0%)	0.24
Peripheral artery obliterans disease, *n* (%)	2 (10.0%)	1 (5.0%)	>0.99
eGFR < 50 mL/min/1.73 m^2^, *n* (%)	4 (20.0%)	1 (5.0%)	0.34
Asthma, *n* (%)	1 (5.0%)	4 (20.0%)	0.34
Malignancy, *n* (%)	2 (10.0%)	1 (5.0%)	>0.99
Autoimmune diseases, *n* (%)	3 (15.0%)	2 (10.0%)	>0.99
**Clinical symptoms in the acute COVID-19 phase**			
Fever, *n* (%)	18 (90.0%)	20 (100.0%)	0.49
Dyspnea, *n* (%)	17 (85.0%)	18 (90.0%)	>0.99
Pleuritic pain, *n* (%)	1 (5.0%)	1 (5.0%)	>0.99
Cough, *n* (%)	15 (75.0%)	18 (90.0%)	0.41
Smell and taste disorders, *n* (%)	3 (15.0%)	7 (35.0%)	0.27
Muscle pain, *n* (%)	2 (10.0%)	5 (25.0%)	0.41
**Parameters at admission to the hospital**			
Systolic blood pressure, mmHg	131.5 (123.5–143.0)	126.5 (120.0–134.5)	0.18
Diastolic blood pressure, mmHg	78.0 (70.0–87.5)	78.5 (69.0–85.0)	0.64
Saturation, %	91 (88–94)	95 (92–96)	**0.043**
Temperature, °C	36.6 (36.6–36.7)	36.6 (36.5–36.6)	0.31
Heart rate, beats per minute	80 (70–110)	87 (76–91)	0.99
Geneva Score, points	4 (1–5)	3 (1–4)	0.24
Pulmonary Embolism Severity Index, points	110 (99–131)	84 (75–97)	**<0.001**
Pack-years, number	0 (0–15)	0 (0–9)	0.85
Systolic blood pressure, mmHg	131.5 (123.5–143.0)	126.5 (120.0–134.5)	0.18
**Treatment pattern before and after admission to the hospital**			
Aspirin, *n* (%)	8 (40.0%)	4 (20.0%)	0.30
Direct oral anticoagulants, *n* (%)	3 (15.0%)	0 (0.0%)	0.23
Beta blockers, *n* (%)	10 (50.0%)	11 (55.0%)	>0.99
Angiotensin II receptor blockers or sartans, *n* (%)	4 (20.0%)	6 (30.0%)	0.72
Statins, *n* (%)	7 (35.0%)	9 (45.0%)	0.52
Diuretics, *n* (%)	4 (20.0%)	7 (35.0%)	0.48
Glucocorticosteroids, orally, *n* (%)	2 (10.0%)	1 (5.0%)	>0.99
Glucocorticosteroids, inhaled, *n* (%)	2 (10.0%)	3 (15.0%)	>0.99
Long-acting beta 2 agonist, *n* (%)	2 (10.0%)	3 (15.0%)	>0.99
Prophylactic LMWH during hospitalization, *n* (%)	3 (15.0%)	2 (10.0%)	>0.99
Intermediate LMWH during hospitalization, *n* (%)	12 (60.0%)	15 (75.0%)	0.31
Therapeutic LMWH, *n* (%)	13 (65.0%)	9 (45.0%)	0.20
Antibiotics during hospitalization, *n* (%)	20 (100.0%)	20 (100.0%)	NA
**Number of hospitalization days**			
Number of hospitalization days	13 (2–21)	22 (18–29)	**0.006**

Categorical variables are presented as numbers with percentages. Continuous variables as medians and Q1–Q3 ranges. Statistically significant differences are in bold. Abbreviations: *n*—number; eGFR—estimated glomerular filtration rate, LMWH—low molecular weight heparin, NA—not applicable.

**Table 7 viruses-17-01400-t007:** Baseline immunological characteristics of severe COVID-19 patients stratified by survival status at follow-up.

Parameter	Deceased COVID-19 Patients in the Follow-Up Among Baseline Severe COVID-19 Cases *n* = 20	Surviving COVID-19 Patients in the Follow-Up Among Baseline Severe COVID-19 Cases *n* = 20	*p*-Value
**Immunological characteristics**			
Lymphocytes CD3+, *n*/µL	348.5 (218.5–547.5)	359.0 (248.0–540.0)	0.65
Lymphocytes CD3+, %	57.7 (48.3–63.2)	55.2 (47.4–61.4)	0.56
Lymphocytes helper CD4+, *n*/µL	167.0 (104.5–352.0)	205.5 (124.0–342.5)	0.65
Lymphocytes helper CD4+, %	31.7 (23.9–41.6)	32.8 (25.6–39.4)	0.87
Lymphocytes cytotoxic CD8+, *n*/µL	109.0 (46.5–141.5)	126.5 (72.0–170.0)	0.40
Lymphocytes cytotoxic CD8+, %	17.9 (12.5–24.8)	16.5 (10.4–26.3)	0.69
Lymphocytes NK, *n*/µL	102.5 (53.0–150.5)	135.5 (60.0–216.5)	0.22
CD4+/CD8+ ratio	1.98 (1.21–3.26)	1.90 (1.40–2.52)	0.71
Lymphocytes B CD19+, *n*/µL	113.5 (56.5–229.0)	114.5 (71.5–208.0)	0.91
Lymphocytes B CD19+, %	20.5 (9.7–29.3)	16.2 (12.9–26.6)	0.85
Immunoglobulin G, g/L	10.4 (9.5–10.7)	10.7 (8.7–12.1)	0.61
Immunoglobulin E, IU/mL	77.2 (31.8–270.5)	49.7 (24.4–242.5)	0.56
**Detailed ratio characteristics**			
Ratio: Lymphocytes B CD19+, %/Lymphocytes helper CD4+, %	0.64 (0.41–0.91)	0.52 (0.42–0.97)	0.87
Ratio: Lymphocytes B CD19+, %/Lymphocytes cytotoxic CD8+, %	1.09 (0.53–1.68)	1.20 (0.68–2.41)	0.77
Ratio: Lymphocytes B CD19+, %/Lymphocytes CD3+, %	0.36 (0.17–0.53)	0.32 (0.25–0.47)	0.98
Ratio: Lymphocytes B CD19+, *n*/µL/Lymphocytes helper CD4+, *n*/µL	0.64 (0.41–0.92)	0.51 (0.42–0.97)	0.87
Ratio: Lymphocytes B CD19+, *n*/µL/Lymphocytes cytotoxic CD8+, *n*/µL	1.11 (0.61–2.32)	1.12 (0.66–1.95)	>0.99
Ratio: Lymphocytes B CD19+, *n*/µL/Lymphocytes CD3+, *n*/µL	0.36 (0.17–0.53)	0.32 (0.25–0.47)	>0.99
Ratio: Lymphocytes B CD19+, *n*/µL/Lymphocytes NK, *n*/µL	1.28 (0.64–1.86)	1.20 (0.62–1.67)	0.71

Continuous variables as medians and Q1–Q3 ranges. Statistically significant differences are in bold. Abbreviations: *n*—number, NK—natural killer, CD—cluster of differentiation.

**Table 8 viruses-17-01400-t008:** Characteristics of COVID-19 survivors from follow-up regarding baseline parameters.

Parameter	Patients with Long-COVID Symptoms *n* = 29	Patients Without Long-COVID Symptoms *n* = 28	*p*-Value
**Demographic characteristics**			
Age, years	53.0 (45.0–63.0)	60.5 (50.8–67.3)	0.14
Sex, male, *n* (%)	20 (69.0%)	17 (60.7%)	0.51
Body mass index, kg/m^2^	29.9 (27.8–31.9)	31.0 (25.8–35.2)	0.55
**Comorbidities**			
Hypertension, *n* (%)	18 (62.1%)	15 (53.6%)	0.52
Diabetes mellitus, *n* (%)	5 (17.2%)	6 (21.4%)	0.69
Hypercholesterolemia, *n* (%)	8 (27.6%)	8 (28.6%)	0.93
Obesity, *n* (%)	10 (34.5%)	11 (39.3%)	0.71
Atrial fibrillation, *n* (%)	2 (6.9%)	1 (3.6%)	>0.99
Chronic heart failure, *n* (%)	4 (13.8%)	2 (7.1%)	0.67
Coronary artery disease, *n* (%)	1 (3.4%)	2 (7.1%)	0.61
Peripheral artery obliterans disease, *n* (%)	5 (17.2%)	2 (7.1%)	0.42
eGFR < 50 mL/min/1.73 m^2^, *n* (%)	2 (6.9%)	0 (0.0%)	0.49
Asthma, *n* (%)	5 (17.2%)	6 (21.4%)	0.69
Malignancy, *n* (%)	0 (0.0%)	2 (7.1%)	0.14
Autoimmune diseases, *n* (%)	3 (10.3%)	4 (14.3%)	0.71
History of recurrent infections, *n* (%)	3 (10.3%)	2 (7.1%)	>0.99
**Clinical symptoms in the acute COVID-19 phase**			
Fever, *n* (%)	24 (82.8%)	20 (71.4%)	0.31
Dyspnea, *n* (%)	23 (79.3%)	22 (78.6%)	0.95
Pleuritic pain, *n* (%)	5 (17.2%)	3 (10.7%)	0.71
Cough, *n* (%)	25 (86.2%)	19 (67.9%)	0.10
Smell and taste disorders, *n* (%)	6 (20.7%)	13 (46.4%)	**0.039**
Muscle pain, *n* (%)	8 (27.6%)	9 (32.1%)	0.71
**Parameters at admission to the hospital**			
Systolic blood pressure, mmHg	129 (120–135)	130 (120–145)	0.75
Diastolic blood pressure, mmHg	80 (70–85)	80 (70–87)	0.91
Saturation, %	95 (94–96)	96 (94–97)	0.26
Temperature, C	36.6 (36.4–36.7)	36.6 (36.5–36.7)	0.64
Heart rate, beats per minute	81 (75–90)	82 (69–92)	0.52
Wells Score, points	0 (0–0)	0 (0–0)	0.50
Geneva Score, points	3 (0–3)	3 (0–4)	0.42
Pulmonary Embolism Severity Index, points	83 (73–94)	90 (79–99)	0.30
Pack-years, number	0 (0–0)	0 (0–2.5)	0.89
**Treatment pattern before and after admission to the hospital**			
Aspirin, *n* (%)	4 (13.8%)	3 (10.7%)	>0.99
Direct oral anticoagulants, *n* (%)	1 (3.4%)	1 (3.6%)	>0.99
Beta blockers, *n* (%)	12 (41.4%)	10 (35.7%)	0.66
Angiotensin II receptor blockers or sartans, *n* (%)	6 (20.7%)	7 (25.0%)	0.70
Statins, *n* (%)	8 (27.6%)	9 (32.1%)	0.71
Diuretics, *n* (%)	8 (27.6%)	10 (35.7%)	0.51
Glucocorticosteroids, orally, *n* (%)	2 (6.9%)	5 (17.9%)	0.25
Glucocorticosteroids, inhaled, *n* (%)	4 (13.8%)	6 (21.4%)	0.50
Long-acting beta 2 agonist, *n* (%)	4 (13.8%)	6 (21.4%)	0.50
Prophylactic LMWH during hospitalization, *n* (%)	11 (37.9%)	10 (35.7%)	0.86
Intermediate LMWH during hospitalization, *n* (%)	13 (44.8%)	15 (53.6%)	0.51
Therapeutic LMWH, *n* (%)	8 (27.6%)	9 (32.1%)	0.71
Antibiotics during hospitalization, *n* (%)	27 (93.1%)	24 (85.7%)	0.42
**Details on hospitalization**			
Number of hospitalization days	19.0 (12.0–22.0)	16 (9–24)	0.74
Severe course of COVID-19, *n* (%)	11 (37.9%)	9 (32.1%)	0.65

Categorical variables are presented as numbers with percentages. Continuous variables as medians and Q1–Q3 ranges. Statistically significant differences are in bold. Abbreviations: *n*—number; eGFR—estimated glomerular filtration rate; LMWH—low molecular weight heparin.

**Table 9 viruses-17-01400-t009:** Results of the survey in COVID-19 patients with and without long-COVID symptoms at the time of 5-year follow-up.

Parameter		COVID-19 Patients in Follow-Up with Long-COVID Symptoms *n* = 29	COVID-19 Patients in Follow-Up Without Long-COVID Symptoms *n* = 28	*p*-Value
Time of recovery, *n* (%)	<3 months	5 (17.2%)	15 (53.6%)	**<0.001**
	3–6 months	8 (27.6%)	10 (35.7%)	
	>6 months	16 (55.2%)	3 (10.7%)	
Persistent headaches after COVID-19, *n* (%)		9 (31.0%)	4 (14.3%)	0.13
Currently headaches after COVID-19, *n* (%)		7 (24.1%)	1 (3.6%)	0.05
Thromboembolic episodes after COVID-19 *, *n* (%)		1 (3.4%)	3 (10.7%)	0.35
Diagnosis of inflammatory joint disease ^$^, *n* (%)		0 (0.0%)	1 (3.6%)	0.49
Diagnosis of connective tissue disease ^#^, *n* (%)		1 (3.4%)	0 (0.0%)	>0.99
Diagnosis of any neurological disease ^@^, *n* (%)		9 (31.0%)	9 (32.1%)	0.93
Diagnosis of new cardiac problems ^&^, *n* (%)		7 (24.1%)	4 (14.3%)	0.34
Diagnosis of renal insufficiency, *n* (%)		4 (13.8%)	0 (0.0%)	0.11
Further episodes of COVID-19, *n* (%)		13 (44.8%)	7 (25.0%)	0.12
Vaccination against COVID-19, *n* (%)		25 (86.2%)	24 (85.7%)	>0.99

Continuous variables are presented as medians and Q1–Q3 ranges. Statistically significant differences are in bold. *—any of the thromboembolic episodes, including deep vein thrombosis, pulmonary embolism, stroke, transient ischemic attack, peripheral artery thrombosis; $—rheumatoid arthritis, spondyloarthropathy, or psoriatic arthritis; #—systemic lupus erythematosus, systemic sclerosis, Sjögren’s disease, or idiopathic inflammatory myopathies; @—symptoms appeared after COVID-19, diseases such as multiple sclerosis, neurodegenerative disease, chronic headaches, insomnia or sleep disorders, depression or other psychiatric disorders; &—coronary heart disease, heart failure, arrhythmias/palpitations, or fainting. Abbreviations: *n*—number; NK—natural killer; CD—cluster of differentiation.

**Table 10 viruses-17-01400-t010:** Comparison of baseline immunological parameters between COVID-19 patients with and without long-COVID symptoms at the time of 5-year follow-up.

Parameter	COVID-19 Patients in Follow-Up with Long-COVID Symptoms *n* = 29	COVID-19 Patients in Follow-Up Without Long-COVID Symptoms *n* = 28	*p*-Value
**Immunological characteristics at baseline**			
Lymphocytes CD3+, *n*/µL	675 (395–889)	623 (410–1031)	0.96
Lymphocytes CD3+, %	62.0 (51.9–71.4)	65.5 (56.0–73.0)	0.65
Lymphocytes helper CD4+, *n*/µL	372 (210–490)	366 (227–512)	0.93
Lymphocytes helper CD4+, %	38.1 (29.9–42.0)	34.4 (27.6–42.6)	0.71
Lymphocytes cytotoxic CD8+, *n*/µL	183 (134–260)	211 (141–275)	0.62
Lymphocytes cytotoxic CD8+, %	14.2 (10.3–23.4)	15.0 (9.0–24.3)	0.78
Lymphocytes NK, *n*/µL	158 (114–236)	156 (78–237)	0.86
CD4+/CD8+ ratio	2.07 (1.40–2.58)	1.80 (1.40–2.25)	0.57
Lymphocytes B CD19+, *n*/µL	132 (89–202)	120 (79.5–185.8)	0.31
Lymphocytes B CD19+, %	16.0 (9.6–19.0)	11.8 (9.5–15.3)	0.11
Immunoglobulin G, g/L	10.8 (9.4–12.1)	11.6 (8.5–13.3)	0.67
Immunoglobulin E, IU/mL	74.9 (24.8–253.0)	50.3 (28.7–212.5)	0.73
**Detailed ratio characteristics**			
Ratio: Lymphocytes B CD19+, %/Lymphocytes helper CD4+, %	0.10 (0.08–0.12)	0.10 (0.07–0.14)	0.77
Ratio: Lymphocytes B CD19+, %/Lymphocytes cytotoxic CD8+, %	1.04 (0.66–1.27)	0.73 (0.42–1.80)	0.48
Ratio: Lymphocytes B CD19+, %/Lymphocytes CD3+, %	0.27 (0.15–0.32)	0.18 (0.13–0.27)	0.11
Ratio: Lymphocytes B CD19+, *n*/µL/Lymphocytes helper CD4+, *n*/µL	0.42 (0.30–0.58)	0.35 (0.24–0.54)	0.25
Ratio: Lymphocytes B CD19+, *n*/µL/Lymphocytes cytotoxic CD8+, *n*/µL	0.86 (0.56–1.24)	0.54 (0.44–0.99)	0.10
Ratio: Lymphocytes B CD19+, *n*/µL/Lymphocytes CD3+, *n*/µL	0.27 (0.15–0.32)	0.18 (0.13–0.27)	0.12
Ratio: Lymphocytes B CD19+, *n*/µL/Lymphocytes NK, *n*/µL	1.04 (0.66–1.27)	0.73 (0.43–1.41)	0.53

Continuous variables are presented as medians and Q1–Q3 ranges. Abbreviations: *n*—number; NK—natural killer; CD—cluster of differentiation.

## Data Availability

The data presented in this study are available on reasonable request from the corresponding author.

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
