# Peer review of "Baseline Dysregulation in B, T, and NK Cells in COVID-19 Predicts Increased Late Mortality but Not Long-COVID Symptoms: Results from a Single-Center Observational Study"

_viruses, 2025, doi:10.3390/v17101400_

Round 1

Reviewer 1 Report

Comments and Suggestions for Authors

Matyja-Bednarczyk et al present work addressing the relationship between early immune dysregulation and long term mortality as well as long COVID symptoms. The work is well structured, with sensible controls, and I really appreciate the sensible way in which the work was conducted. The long follow up period of 54 months is a key strength. There is also a large panel of immunological data coupled with well-run statistics. Many Long COVID studies are sensational and not well reported, but this one is very well conducted. I only have minor textual issues.

  1. Long COVID Definition
    • What definition of Long COVID was used? It would help in the methods to explain the assessment and the paper would benefit from aligning with WHO’s or NICE’s definitions explicitly.
    • There is a reliance on retrospective, self reported symptoms via telephone surveys which  introduce recall bias, can the authors mention this in their work.
    • Some work on Long COVID has focussed more on single symptoms such as linking severe fatigue alone to IFN-y secretion. It may be that the authors see fewer correlations because of the general nature of the symptoms assessed. I.e. that Long COVID is lilkely multiple syndromes all under a single umbrella term. The authors should mention this or similar in their discussion.
  2. Do the authors use multivariate analysis at all to control for age, sex, and major comorbidities when assessing mortality predictors?
  3. The authors could also provide effect sizes and confidence intervals for key associations?
  4.  Line 368: there are other works looking at immune cell balance in Long COVID including: doi= 10.1126/sciadv.adi9379 (higher monocytes, lower CD4s), https://www.cell.com/immunity/fulltext/S1074-7613(22)00046-2 (increased B cell numbers), https://pmc.ncbi.nlm.nih.gov/articles/PMC10620090/ (higher CD14+CD16- monocytes, and HLA-DR+ B cells).
  5. It is quite interesting to note that the authors saw no differences in vaccination and long covid symptoms. This is an interesting finding that it not discussed as far as I can tell. There’s quite some literature on this topic already. The authors may wish to interpret the field differently but in my opinion,
    • I tend to see that vaccination reduces some symptoms of Long COVID in some studies but less in others. Vaccines seem to reduce the risk of severe fatigue (doi=10.1093/cid/ciac630) 6 months most infection and many other symptoms weeks after infection (10.1038/s41541-022-00526-5, 10.1093/ofid/ofac464). However the risk of post-COVID cardiovascular events is not reduced by much (10.1038/s41591-022-01840-0). Could the authors discuss their findings within the framework of these results?  

Author Response

Reviewer 1

Matyja-Bednarczyk et al present work addressing the relationship between early immune dysregulation and long term mortality as well as long COVID symptoms. The work is well structured, with sensible controls, and I really appreciate the sensible way in which the work was conducted. The long follow up period of 54 months is a key strength. There is also a large panel of immunological data coupled with well-run statistics. Many Long COVID studies are sensational and not well reported, but this one is very well conducted. I only have minor textual issues.

General response to comments:

We sincerely thank Reviewer 1 for the careful evaluation of our work and for providing thoughtful suggestions, which have greatly helped us improve the manuscript. Below, we respond to each comment in detail.

  1. Long COVID Definition

What definition of Long COVID was used? It would help in the methods to explain the assessment and the paper would benefit from aligning with WHO’s or NICE’s definitions explicitly.

Response:

We thank the Reviewer for pointing this out. In the Methods section, we have clarified that, in this study, we adopted the WHO definition of long-COVID syndrome, which refers to symptoms that begin at least three months after infection and persist for a minimum of two months.

There is a reliance on retrospective, self reported symptoms via telephone surveys which introduce recall bias, can the authors mention this in their work.

Response:

We thank the Reviewer for raising this important point. We agree that reliance on retrospective, self-reported symptoms collected through telephone surveys may introduce recall bias. We have now acknowledged this limitation in the revised Discussion section.

Some work on Long COVID has focussed more on single symptoms such as linking severe fatigue alone to IFN-y secretion. It may be that the authors see fewer correlations because of the general nature of the symptoms assessed. I.e. that Long COVID is lilkely multiple syndromes all under a single umbrella term. The authors should mention this or similar in their discussion.

Response:

We thank the Reviewer for this insightful comment. We agree that long-COVID is a heterogeneous condition likely comprising multiple syndromes, and that the use of broad symptom categories may have limited the strength of observed correlations. We have now addressed this point in the Discussion section.

  1. Do the authors use multivariate analysis at all to control for age, sex, and major comorbidities when assessing mortality predictors?

Response:

We thank the Reviewer for this valuable comment. We have recalculated the main outcomes (Tables 1-3), adjusting for age, sex, and body mass index. We have also clarified this approach in the revised Statistical Analysis section. However, the main conclusions remained unchanged.

  1. The authors could also provide effect sizes and confidence intervals for key associations?

Response:

We thank the Reviewer for this suggestion. We agree that the paper does not include effect measures. However, for us, the statistical significance of the effect itself was more important than assessing its strength. This is especially true given that the literature, depending on the coefficients, provides various classifications of the effect as weak, moderate, or strong, depending on the effect measurements. We commented on that issue in the improved manuscript’s version.

  1. Line 368: there are other works looking at immune cell balance in Long COVID including: doi= 10.1126/sciadv.adi9379 (higher monocytes, lower CD4s), https://www.cell.com/immunity/fulltext/S1074-7613(22)00046-2 (increased B cell numbers), https://pmc.ncbi.nlm.nih.gov/articles/PMC10620090/ (higher CD14+CD16- monocytes, and HLA-DR+ B cells).

Response:

We thank the Reviewer for suggesting these relevant studies. We have incorporated and discussed the findings from all of the mentioned papers in the revised Discussion section.

  1. It is quite interesting to note that the authors saw no differences in vaccination and long covid symptoms. This is an interesting finding that it not discussed as far as I can tell. There’s quite some literature on this topic already. The authors may wish to interpret the field differently but in my opinion, I tend to see that vaccination reduces some symptoms of Long COVID in some studies but less in others. Vaccines seem to reduce the risk of severe fatigue (doi=10.1093/cid/ciac630) 6 months most infection and many other symptoms weeks after infection (10.1038/s41541-022-00526-5, 10.1093/ofid/ofac464). However the risk of post-COVID cardiovascular events is not reduced by much (10.1038/s41591-022-01840-0). Could the authors discuss their findings within the framework of these results?

Response:

We thank the Reviewer for highlighting this important point. However, our study was performed on COVID-19 patients between June and November 2020, that is, before the vaccination was available. Thus, neither of our patients was vaccinated before the first case of COVID-19. That is likely why vaccination status has no impact on the frequency of long-COVID symptoms in our cohort. We have now discussed this issue in the revised Discussion section and incorporated the suggested studies to place our results in the context of the existing literature.

We hope that the revised version of the manuscript will now be considered suitable for publication. Once again, we sincerely thank you for your valuable comments and constructive suggestions.

Reviewer 2 Report

Comments and Suggestions for Authors

It is an interesting study, as you stated it uses a relatively small sample size from a particular area, hard to generalize to the overall Polish population

On vaccination, a reader may wonder if different boosters of vaccination was taken into account? Prior history of vaccinations? I know you assessed current COVID vaccination, what about past?

On comorbidities, did you look at upper respiratory infections (e.g., influenza, pneumonia, rhinovirus, etc?)

Author Response

Reviewer 2

It is an interesting study, as you stated it uses a relatively small sample size from a particular area, hard to generalize to the overall Polish population

General response to comments:

We sincerely thank Reviewer 2 for the careful evaluation of our manuscript and for providing constructive feedback. Below, we address each of the comments in detail.

On vaccination, a reader may wonder if different boosters of vaccination was taken into account? Prior history of vaccinations? I know you assessed current COVID vaccination, what about past?

Response:

We thank the Reviewer for raising this important point. However, our study was performed on COVID-19 patients between June and November 2020, when vaccines were not yet available. That is likely why vaccination status has no impact on our data. Furthermore, the vaccination status in the follow-up analysis was based on a retrospective analysis, resulting in difficulty in recalling detailed vaccination histories and made it unfeasible to obtain reliable data. We have now acknowledged this limitation in the Discussion section.

On comorbidities, did you look at upper respiratory infections (e.g., influenza, pneumonia, rhinovirus, etc?)

Response:

We acknowledge that this is an important aspect of our study. In follow-up, patients were asked about the frequency of respiratory tract infections. However, a retrospective analysis made it infeasible to obtain reliable data, preventing us from drawing firm conclusions. We have noted this as a limitation in the Discussion section.

We hope that the revised version of the manuscript will now be considered suitable for publication. Once again, we sincerely thank you for your valuable comments and constructive suggestions.

Reviewer 3 Report

Comments and Suggestions for Authors

The manuscript investigates immunological, clinical, and laboratory characteristics of patients with severe and mild COVID-19 and examines long-term outcomes, including mortality and long-COVID symptoms. The study provides detailed data on lymphocyte subsets, inflammatory markers, and their association with five-year mortality, contributing to the understanding of COVID-19 pathophysiology and prognosis. Several aspects, however, require improvement to enhance clarity, methodological rigor, and reproducibility.

The title is informative but could be more concise and highlight the study’s novelty and principal findings. The abstract provides a general overview but would benefit from a structured format distinguishing Background, Methods, Results, and Conclusions, and should include quantitative results (e.g., effect sizes, p-values) to strengthen its impact. Some keywords are overly broad and could be refined to improve discoverability.

The introduction presents relevant background information but does not fully emphasize the research gap or the unique contribution of the study. Condensing descriptive content and integrating recent literature would improve context and relevance.

The Methods section is generally adequate; however, additional detail is needed regarding inclusion and exclusion criteria, statistical analyses (specific tests, software versions, adjustments for multiple comparisons), handling of missing data, and ethical approvals.

Results are supported by quantitative data in tables, including precise p-values and relevant measures; nevertheless, figure legends and in-text descriptions could better highlight these values to improve clarity and interpretation.

The Discussion provides appropriate interpretation of the findings but occasionally overstates conclusions. Statements should be evidence-based and moderated. Comparisons with prior studies should be strengthened, highlighting consistencies and discrepancies. Expanding on limitations—including sample size, follow-up attrition, single-center design, and generalizability—would enhance rigor. A concise discussion of potential clinical or translational implications would also be valuable. Minor grammatical and stylistic revisions, consistent abbreviation usage, and reference formatting adjustments would improve overall readability and presentation.

Addressing these points would enhance the scientific rigor, clarity, and impact of the manuscript.

Author Response

Reviewer 3

The manuscript investigates immunological, clinical, and laboratory characteristics of patients with severe and mild COVID-19 and examines long-term outcomes, including mortality and long-COVID symptoms. The study provides detailed data on lymphocyte subsets, inflammatory markers, and their association with five-year mortality, contributing to the understanding of COVID-19 pathophysiology and prognosis. Several aspects, however, require improvement to enhance clarity, methodological rigor, and reproducibility.

General response to comments:

We thank Reviewer 3 for the thoughtful evaluation of our work. The detailed remarks and recommendations helped us to improve the manuscript. Below, we provide point-by-point responses to each comment.

The title is informative but could be more concise and highlight the study’s novelty and principal findings. The abstract provides a general overview but would benefit from a structured format distinguishing Background, Methods, Results, and Conclusions, and should include quantitative results (e.g., effect sizes, p-values) to strengthen its impact. Some keywords are overly broad and could be refined to improve discoverability.

Response:

We thank the Reviewer for this valuable feedback. The title has been revised to emphasize the principal findings better while avoiding redundancy. The abstract has been restructured in line with the journal’s requirements, which do not allow explicit subheadings such as Background, Methods, Results, and Conclusions; instead, the text has been organized accordingly. We have also refined the presentation of the results by including p-values and other quantitative data, and the keywords have been revised for greater specificity.

The introduction presents relevant background information but does not fully emphasize the research gap or the unique contribution of the study. Condensing descriptive content and integrating recent literature would improve context and relevance.

Response:

We thank the Reviewer for this helpful comment. The Introduction section has been revised to more clearly emphasize the research gap and the unique contribution of our study.

The Methods section is generally adequate; however, additional detail is needed regarding inclusion and exclusion criteria, statistical analyses (specific tests, software versions, adjustments for multiple comparisons), handling of missing data, and ethical approvals.

Response:

We thank the Reviewer for this helpful comment. The Methods section has been revised to include more detail on inclusion and exclusion criteria, statistical analyses, handling of missing data, and ethical approvals.

Results are supported by quantitative data in tables, including precise p-values and relevant measures; nevertheless, figure legends and in-text descriptions could better highlight these values to improve clarity and interpretation.

Response:

We acknowledge that a more detailed description of the results would improve readability; therefore, we have expanded the in-text descriptions and, where appropriate, included references to the more comprehensive results presented in the tables.

The Discussion provides appropriate interpretation of the findings but occasionally overstates conclusions. Statements should be evidence-based and moderated. Comparisons with prior studies should be strengthened, highlighting consistencies and discrepancies. Expanding on limitations—including sample size, follow-up attrition, single-center design, and generalizability—would enhance rigor. A concise discussion of potential clinical or translational implications would also be valuable. Minor grammatical and stylistic revisions, consistent abbreviation usage, and reference formatting adjustments would improve overall readability and presentation.

Response:

We thank the Reviewer for this insightful comment. The Discussion section has been revised to temper any potentially misleading statements and to expand the comparison with previous studies. We have also elaborated on the limitations of our study and emphasized its clinical relevance and potential translational implications. Furthermore, the text has been carefully revised for grammar, style, and formatting.

Addressing these points would enhance the scientific rigor, clarity, and impact of the manuscript.

Response:

We hope that the revised version of the manuscript will now be considered suitable for publication. Once again, we sincerely thank you for your valuable comments and constructive suggestions.